# Complete Genome and Characterization Analysis of a *Bifidobacterium animalis* Strain Isolated from Wild Pigs (*Sus scrofa ussuricus*)

**DOI:** 10.3390/microorganisms13071666

**Published:** 2025-07-16

**Authors:** Tenggang Di, Huan Zhang, Cheng Zhang, Liming Tian, Menghan Chang, Wei Han, Ruiming Qiao, Ming Li, Shuhong Zhang, Guangli Yang

**Affiliations:** 1College of Biology and Food, Shangqiu Normal University, Shangqiu 476000, China; 15083129326@163.com (T.D.); zhanghuan231101@163.com (H.Z.); zhangshu9281@163.com (C.Z.); liming202506@163.com (L.T.); changmenghan1005@163.com (M.C.); hanwei00005@163.com (W.H.); 2College of Animal Science and Technology, Henan Agricultural University, Zhengzhou 450046, China; qrm480@163.com (R.Q.); liming@henau.edu.cn (M.L.); 3Gansu Key Laboratory of Herbivorous Animal Biotechnology, Faculty of Animal Science and Technology, Gansu Agricultural University, Lanzhou 730070, China

**Keywords:** wild pigs, isolation, *B. animalis* strain, genomic characterization, protective mechanism

## Abstract

*Bifidobacterium* is a predominant probiotic in animals that is associated with host intestinal health. The protective mechanisms of the *Bifidobacterium animalis* (*B. animalis*) strain, specifically those related to functional gene–host interactions in intestinal homeostasis, remain poorly elucidated. This study reports the complete genome sequence and characterization of a *B. animalis* strain isolated from wild pig feces, which comprised a single circular chromosome (1,944,022 bp; GC content 60.49%) with 1567 protein-coding genes, and the *B. animalis* strain had certain acid resistance, bile salt resistance, gastrointestinal fluid tolerance, and antibacterial characteristics. Genomic annotation revealed three putative genomic islands and two CRISPR-Cas systems. Functional characterization identified genes encoding carbohydrate-active enzymes (CAZymes) and associated metabolic pathways, indicating that this strain can degrade complex dietary carbohydrates and synthesize bioactive metabolites for gut homeostasis. Although the antibiotic resistance genes were predicted, phenotypic assays demonstrated discordant resistance patterns, indicating complex regulatory networks. This study indicated the genomic basis of *Bifidobacterium*–host crosstalk in intestinal protection, providing a framework for developing novel probiotic interventions.

## 1. Introduction

An animal gut has trillions of microbial inhabitants, which form a complex symbiotic ecosystem with the host. Evidence indicates that gut microorganisms profoundly regulate host health, metabolic processes, and systemic immunity [1,2,3,4]. Therefore, investigating the composition and functional dynamics of gut microorganisms will enhance our understanding of microbial resources and host–microbe interactions. Studies on wild pig gut microorganisms have revealed distinct compositional differences compared to domestic and commercial pigs. Furthermore, wild pigs indicated higher abundances of cellulose-degrading bacteria (e.g., *Ruminococcus* spp.) and short-chain fatty acid-producing taxa (e.g., *Prevotella* spp.), as well as lower pathogen loads and greater ecological stability [5,6]. The domestic and commercial pigs had enriched populations of *Streptococcus* and *Lactobacilli*, compared to *Bifidobacterium*, which predominates in their wild counterparts [7]. This divergence in microbial profiles, particularly the depletion of co-evolved beneficial microbes, may underline the improved environmental adaptability and disease resistance observed in wild pigs compared to domesticated breeds.

In modern livestock production, routine antibiotic administration has been employed to enhance growth performance and prevent infectious diseases. However, this practice may disrupt gut microorganism assembly and raise food safety concerns [8,9]. Wild pigs are not exposed to intensive farming and, therefore, have indicated significantly lower resistance abundance compared to commercial pigs [10]. In swine production, the use of antibiotic growth promoters is being restricted [9]; thus, developing antibiotic alternatives has become imperative to maintaining post-weaning piglet health. Comparative analyses of wild and domestic porcine microbiome have indicated that harnessing wild pig-derived probiotics through isolation and transplantation could restore gut eubiosis and enhance host resilience [11]. Studies suggest that probiotics are viable antibiotic substitutes as they modulate microbial composition and suppress pathogen colonization [12].

Over the past decade, probiotic feed additives have emerged as a research priority in animal husbandry. Commercially utilized strains primarily include *Bifidobacterium*, *Lactobacilli*, *Streptococcus*, and yeast species [13]. Among these, *Bifidobacterium* has garnered extensive attention due to its functional efficacy. This Gram-positive, non-motile, and strictly anaerobic genus colonizes multiple ecological niches, including the gastrointestinal tract and oral cavity [14,15,16]. It is predominantly found in breastfed infant guts, where it utilizes human milk oligosaccharides for early colonization, enhancing intestinal barrier integrity and developmental health [17,18,19,20,21]. Experimental models demonstrate that *Bifidobacterium* optimizes gut ecology and exerts immunomodulatory effects [22,23,24,25]. Furthermore, mechanistic studies have revealed its capacity to fortify intestinal barrier function [26,27,28,29,30,31,32] and mitigate enteric inflammation through prophylactic administration [33]. Moreover, *Bifidobacterium animalis* (*B. animalis*) supplementation has been found to improve growth metrics, enrich beneficial taxa, and suppress pathogens; its precise immune regulation and barrier maintenance mechanisms warrant further investigation [12].

Although *Bifidobacterium*’s has several health-promoting effects, the genome landscapes of the *B. animalis* remain incompletely characterized. The mechanistic interplay between their genome architecture and host interactions in intestinal protection lacks systematic elucidation. This study isolated a *B. animalis* strain from wild pig feces and performed whole genome sequencing to delineate its genome features, thus establishing molecular foundations for investigating gut inflammatory regulation. This study aims to elucidate the anti-pathogenic mechanisms and antimicrobial resistance traits of this strain, offering a theoretical framework for developing next-generation probiotic formulations.

## 2. Materials and Methods

### 2.1. Isolation of the B. animalis Strain

In this study, three healthy female adult wild pigs (4-year-old) were selected and maintained on a diet primarily composed of corn and soybeans, supplemented with hay, with twice-daily feeding [7]. Animals were provided with ad libitum access to water and received no antibiotic treatment throughout the study period [7]. Fresh feces were sampled and immediately transferred into an anaerobic glove box (Electrotek, Aberdeen, UK), serially diluted with PBS (pH 7.0 sterilized), and then inoculated on a selective Neomycin–Bacitracin–Nalidixic Acid–Lithium Chloride (NPNL) medium at 37 °C for 48 h under aerobic conditions. A single colony of the strains was repeatedly picked and streaked on fresh selective NPNL medium to obtain pure colonies. The isolated colonies were then inoculated in the corresponding TPY medium and stored at −80 °C in the broths augmented with 20% glycerol.

### 2.2. Microscopic Analyses of the B. animalis Strain

Morphological characterization included Gram staining and scanning electron microscopy (SEM). The isolated and purified bacterial strain underwent Gram staining and was examined using phase contrast microscopy (100×) to check for color changes and determine Gram positivity. For electron microscopic analysis, bacterial cells were fixed, dehydrated, embedded, sectioned, and stained with heavy metal salts. High-resolution images of cellular morphology and structure were obtained using a transmission electron microscope.

### 2.3. DNA Extraction and Polymerase Chain Reaction (PCR)

Strain identification was verified through 16S rRNA gene sequencing before DNA isolation. Each isolate was then subjected to PCR amplification followed by sequencing. To optimize the annealing temperature, a temperature gradient PCR (12 points from 53 °C to 63 °C) was performed before amplification. Full-length 16S rRNA genes were amplified using primers 27F (5′-AGAGTTTGATCCTGGCCTCAG-3′) and 1492R (5′-GGTTACCTTGTTACGACTT-3′). The PCR protocol included initial denaturation at 98 °C for 3 min; 35 cycles at 98 °C for 30 s (denaturation), 56.1 °C for 30 s (annealing), and 72 °C for 1 min (extension); and a final extension at 72 °C for 10 min. Purified PCR products were sequenced via Sanger sequencing, and strain identity was confirmed by BLAST (Version 2.3.0) alignment against genomic databases. MEGA 6.0 software was employed to construct the phylogenetic tree using the Neighbor-Joining (NJ) method [34]. Genomic DNA was extracted using the TIANGEN Genomic DNA Purification Kit (DP302; Tiangen Biotech Co., Ltd., Beijing, China) as per the manufacturer’s protocol. DNA quality and concentration were assessed by 1% agarose gel electrophoresis and NanoDrop Microvolume Spectrophotometer (Thermo Fisher Scientific, Waltham, MA, USA), respectively.

### 2.4. Characterization of the B. animalis Strain

The *B. animalis* strain was separated and cultured with selective NPNL medium and then inoculated into TPY medium for drawing the growth curve, acid resistance, bile salt resistance, and gastrointestinal fluid tolerance assays. The growth kinetics were determined by measuring the OD value at 600 nm using an ultraviolet spectrophotometer, and the obtained values were used to plot the growth curve. The acid resistance test was to adjust the pH value of TPY medium to 1.5, 2.5, 3.5, and 4.5 with 1 mol/L hydrochloric acid, and the activated different *B. animalis* strain suspension was inoculated at 10% (*v*/*v*). After 3 h of culture, the *B. animalis* strain solution was taken out for gradient dilution and coated on the surface of TPY medium, and the number of viable *B. animalis* strains in different pH culture medium was calculated after 48 h. The bile salt assay was to inoculate 10% suspension of the *B. animalis* strain into TPY medium with 0%, 0.1%, 0.2%, 0.3%, and 0.4% concentrations of swine bile salt (S3895, Shanghai, China), and then cultured under anaerobic conditions at 37 °C. After 1 h and 12 h, the number of viable *B. animalis* strain was calculated at different bile salt concentrations. The gastrointestinal fluid tolerance experiment is to shake and mix the solution of the *B. animalis* strain evenly into a suspended state, take 1 mL and inoculate it into 9 mL of artificially prepared stomach fluid with a pH of 3.0 (take 1 mol/mL of hydrochloric, add water to dilute it, adjust the pH to 1.5, and add 1 g of pepsin to each 100 mL) or artificial intestinal fluid with a pH of 8.0 (add 1 g of trypsin to each 100 mL of liquid), mix well and under anaerobic conditions at 37 °C for 3 h, then take out the *B. animalis* strain solution for gradient dilution, apply it to the surface of the TPY medium, and calculate the number of live *B. animalis* strains after 48 h.

For the Oxford cup method for the detection of the *B. animalis* in vitro antibacterial assay, the activated *B. animalis* was centrifuged at 4 °C, 5000 r/min for10 min, and then filtered with a small filter (0.22 μm) for sterilization. The activated *Escherichia coli* K88 was then evenly added to the cooled nutrient broth agar at a ratio of 1:100, and 15–20 mL was poured into the plate. It was placed in a sterile environment for 30 min to be quickly evaporate the excess water. Sterile forceps were used to pick up the Oxford cup and quickly pass the flame of the alcohol lamp, and the sample was then placed in the dish. The nutrient broth culture solution supernatant was added to the control and the experimental group, respectively. After the Oxford cup was filled, it was placed at room temperature 4–6 h to allow it to diffuse completely. It was placed in a 37 °C incubator for a constant temperature culture for 24 h, and the inhibitory circle was observed. If one appeared, it indicated an inhibitory effect, and the diameter of the inhibitory circle was measured.

### 2.5. Whole Genome Sequencing, Quality Control, and Assembly

High-quality DNA samples were subjected to de novo sequencing by Meiji Biomedical Technology Co., Ltd., (Shanghai, China). PacBIo RS II (20 kb SMRTbellTM templates) and Illumian HiSeq4000 (TruSeq DNA PCR- Free 350 bp library) were employed for the whole genome sequencing of the isolated *B. animalis* strain. The bioinformatics workflow integrated sequencing data from PacBio RS II and Illumina platforms. First, quality filtering was performed on raw FASTQ data to remove reads with quality scores below Q20 that were shorter than 25 bp after trimming, contained >10% ambiguous bases (N), and generated high-confidence clean datasets. Then, PacBio long reads were assembled into contigs using Unicycler v0.4.8, and sequence correction was performed using Pilon v1.22 to determine the starting point of the circular genome. The genome map was generated using Circos software (Version 0.69-6) [35].

### 2.6. Genome Composition Analysis and Functional Annotation

The genome’s coding sequences (CDS) were predicted using Prodigal v2.6.3 [36], and plasmid genes were predicted using GeneMarkS (Version 4.3) [37]. Non-coding RNAs, tRNAs, were predicted using tRNAscan-SE v2.0 [38], and rRNAs were predicted using Barrnap (Version 0.9) [39]. Genomic islands were assessed via the Island Viewer (Version 1.0.0) online system, while CRISPR-Cas systems were detected using CRISPRCasFinder (Version 3) [40]. Functional annotation and classification of predicted genes were performed using tools such as Diamond (Version 0.8.35), HMMER (Version 3.1b2), and BLASTP (Version 2.3.0) [41], including Kyoto Encyclopedia of Genes and Genomes (KEGG), Virulence Factors Database (VFDB), Carbohydrate-Active Enzymes (CAZymes), Clusters of Orthologous Groups (COG), Pathogen–Host Interactions (PHI), NCBI Non-Redundant Protein Sequence Database (NR), Gene Ontology (GO), UniProt/Swiss-Prot, Pfam, and the Comprehensive Antibiotic Research Database (CARD).

## 3. Results

### 3.1. Isolation and Identification of the B. animalis Strain

This study isolated a *B. animalis* strain from fresh fecal samples of healthy wild pigs under anaerobic conditions using a selective NPNL medium. Colony morphology analysis revealed circular, moist, convex, white, and translucent colonies with distinct margins (Figure 1A,B). Gram staining confirmed the strain was Gram-positive with rod-shaped or short rod-shaped morphology and occasional bifurcated termini (Figure 1C,D). Scanning electron microscopy further characterized the cells as short rods arranged in pairs or clusters, aligning with typical *Bifidobacterium* morphology (Figure 1E). This analysis result revealed unique surface structures that may be associated with host adhesion and environmental adaptation. To validate taxonomic classification, single colonies were also purified through repeated streaking and subjected to full-length 16S rRNA gene sequencing for definitive identification (Appendix A).

### 3.2. Characteristics of the B. animalis Strain

Growth kinetic profiling displayed a classical growth curve: a robust logarithmic growth phase initiated at 4 h, reached a plateau phase by 18 h, followed by a decline phase commencing at 22 h (Figure 2A). The physiological characteristics results showed that the *B. animalis* strain had good acid resistance, and the survival rate of the *B. animalis* strain has reached 82.3% at pH 3.5. In terms of bile salt tolerance, the survival rate of the *B. animalis* strain reached more than 100% when cultured at 0.2% bile salt concentration for 12 h. The survival rate of the *B. animalis* strain cultured in gastrointestinal fluid reached more than 65%, of which the survival rate of the *B. animalis* strain reached 190.74% and 65.4% after being cultured in artificial gastric juice and intestinal juice, respectively. The results of the in vitro antibacterial test of the *B. animalis* strain against *E. coli* K88 by the Oxford cup method showed that the diameter of the inhibition circle was between 10 and 15 mm (Figure 2B). Therefore, the *B. animalis* strain has a significantly stronger inhibitory effect on enterotigenic *E. coli* K88. These results provide a theoretical basis for the development, utilization, and production of green probiotics as a substitute for antibiotic products, based on the microbial resources of the *B. animalis* strain in the gut microbiota of wild pigs.

### 3.3. Genomic Information of the B. animalis Strain

To characterize the genome architecture of the isolated *B. animalis* strain, a hybrid sequencing strategy combining next-generation sequencing (Illumina) and third-generation sequencing (PacBio, Menlo Park, CA, USA) was employed. The complete genome assembly revealed a single circular chromosome of 1,944,022 bp with a GC content of 60.49%. Taxonomic classification was performed using whole-genome annotation against the NCBI RefSeq database. The genome encoded 1567 protein-coding genes, 51 tRNA genes, 12 rRNA genes, and 9 sRNA genes, with the circular genome map illustrated in Figure 3. The outermost circle is the identification of genome size. The second and the third circle are the coding DNA sequences (CDS) on the positive and negative strands, respectively, and different colors indicate the different functional annotations of CDS in the clusters of orthologous groups (COGs) database. The fourth circle is rRNA and tRNA. The fifth circle is the GC content. The innermost circle is the GC Skew value, and its algorithm is G-C/G+C, which can assist to determine the leading strand and lagging strand. The legend circle1 is the functional classification in the COG database, and the legend circle2 is a different RNA classification. Genomic island analysis via Island Viewer identified three islands (ranging from 8012 to 16,760 bp) located on the chromosome (Appendix A). Detailed annotations are provided in Appendix A. Furthermore, two distinct CRISPR-Cas systems were identified through spacer sequence analysis, which showed no homology to known phage sequences.

### 3.4. Functional Annotation

The COG annotation annotated 1297 genes into 23 gene types, which accounti for 82.77% of the genes in the *B. animalis* strain (Figure 4). The number of each gene type were as follows: 87 K-type genes (transcription), 83 L-type genes (replication, recombination and repair), 6 N-type genes (cell motility), 44 C-type genes (energy production and conversion), 145 G-type genes (carbohydrate transport and metabolism), 1 A-type gene (RNA processing and modification), 83 R-type genes (general function prediction only), 171 J-type genes (translation, ribosomal structure, and biogenesis), 62 P-type genes (inorganic ion transport and metabolism), 3 W-type genes (extracellular structures), 97 M-type genes (cell wall/membrane/envelope biogenesis), 48 I-type genes (lipid transport and metabolism), 68 O-type genes (posttranslational modification, protein turnover, chaperones), 73 F-type genes (nucleotide transport and metabolism), 8 Q-type genes (secondary metabolites biosynthesis, transport, and catabolism), 38 S-type genes (function unknown), 73 H-type genes (coenzyme transport and metabolism), 60 T-type genes (signal transduction mechanisms), 162 E-type genes (amino acid transport and metabolism), 50 V-type genes (defense mechanisms), 27 D-type genes (cell cycle control, cell division, chromosome partitioning), 15 U-type genes (intracellular trafficking, vesicular transport, and secretion), and 12 X-type genes (mobilome: transposons, prophages). In these distributions, the most abundant categories were translation, biogenesis, and ribosomal structure (171), amino acid metabolism and transport (162), and carbohydrate metabolism and transport (145). All COG functional annotations are enlisted in Appendix A.

Gene Ontology (GO) functional annotation classified 1280 genes (81.68% of the total) into three categories: cellular component (CC), molecular function (MF), and biological process (BP) (Appendix A). Among these, 629 genes were related to CC, 1047 to MF, and 670 to BP. Within the BP category, the most enriched terms were translation (62 genes, 9.25%) and carbohydrate metabolism (39 genes, 5.82%). For CC, the predominant annotations included integral membrane components (335 genes), cytoplasm (174 genes), and plasma membrane (110 genes). In the MF category, the highest enrichment was observed in ATP binding (214 genes), followed by DNA binding (119 genes), hydrolase activity (114 genes), and metal ion binding (78 genes). A comprehensive visualization of these distributions is provided in Figure 5. Complete GO annotation data are available in Appendix A.

KEGG pathway analysis identified 1222 genes annotated into 38 metabolic pathways (Figure 6). The complete annotation data is provided in Appendix A. The most enriched pathways included carbohydrate metabolism (107 genes), amino acid metabolism (105 genes), nucleotide metabolism (55 genes), and cofactor/vitamin metabolism (54 genes). Cluster analysis grouped 38 pathways into six major categories: cellular processes (72 genes), human diseases (104 genes), metabolism (842 genes), organismal systems (28 genes), genetic information processing (142 genes), and environmental information processing (86 genes). Furthermore, 12 subcategories were identified within the metabolism category (842 genes), with the highest gene counts observed in global/overview maps (339 genes), carbohydrate metabolism (107 genes), and amino acid metabolism (105 genes). In the cellular processes category (72 genes), the highest gene counts were observed in the cellular community-prokaryotes (46 genes) and cell growth/death (17 genes) subcategories. For genetic information processing (142 genes), subcategories were dominated by translation (79 genes), replication/repair (38 genes), and folding/sorting/degradation (21 genes). In the human diseases category (104 genes), the most prominent subcategories included antimicrobial drug resistance (12 genes), cancer drug resistance overview (9 genes), and bacterial infectious diseases (8 genes). The organismal systems category (28 genes) showed enrichment in the endocrine system (11 genes), aging (5 genes), environmental adaptation (5 genes), and the digestive system (4 genes). Finally, environmental information processing (86 genes) comprised membrane transport (61 genes) and signal transduction (25 genes).

### 3.5. Prediction of Carbohydrate-Active Enzyme Genes

This study also evaluated the characteristics of the carbohydrate-active-enzymes-encoding genes in the *B. animalis* strain’s genome, which revealed 78 genes coding the putative CAZymes (Appendix A, Figure 7). Based on the CAZy enzymatic classification, the genes comprising carbohydrate-active domains were categorized into 5 classes, including auxiliary activities enzymes (1), carbohydrate-binding modules (CBMs, 2), carbohydrate esterases (CEs, 12), glycosyl transferases (GTs, 25), and glycoside hydrolases (GHs, 38). Of all the CAZyme-associated genes, the most abundant were GTs, GHs, and CEs, which were linked with carbohydrate-degrading enzymes, accounting for 96.15% of all the CAZymes-encoding genes. Furthermore, the clusters of polysaccharide utilization genes were substantially diverse and characterized by different combinations of specific CAZy families and proteins of unknown activities. The degradation enzymes most frequently encoded in the gene clusters of polysaccharide utilization included β-xylosidases (GH43), β-glucosidases (GH1, GH3), β-galactosidases (GH2, GH42, GH36), and α-amylases (GH13). These were involved in the breakdown of cellulose, oligosaccharides, and starch. The *B. animalis* strain genome indicated substantial enrichment of GH28 and GH32 family genes within polysaccharide utilization loci (PULs), critical for hydrolyzing hemicellulosic polysaccharides. Furthermore, GT2 (glycosyltransferase family 2) was identified as the most abundant glycosyltransferase family among the eight distinct glycosyltransferase families. Carbohydrate esterases (CEs) were observed to cleave ester-linked groups such as arylesterase (*PON2*) and acetyl xylan esterase (*AxE*) from carbohydrate substrates. Although carbohydrate-binding modules (CBMs) lack intrinsic hydrolytic activity, they enhance catalytic efficiency by mediating interactions between CAZymes and carbohydrate ligands. These findings revealed that GH family genes in the *B. animalis* strain genome are pivotal for degrading plant-derived polysaccharides, underscoring the strain’s potential as a probiotic candidate.

### 3.6. Prediction of Virulence Genes

*B. animalis* strain is a widely used probiotic species and has significant functional heterogeneity across strains, particularly in the biological roles of their putative virulence genes. In this study, genomic analysis identified 139 putative virulence genes in a *B. animalis* strain (Appendix A, Figure 8), with 104 functionally annotated in the VFDB. These comprised 39 offensive, 23 defensive, 7 regulatory, and 35 nonspecific virulence-associated genes. The detection of these genes does not directly confer pathogenic potential. Probiotic bacteria may utilize such genes to adaptively function through host interactions or intestinal niche competition. Of the 39 offensive virulence genes, 16 were adhesion-associated, including motility-related genes, flagellar assembly components, *Flp type IV pili* formation genes, and adhesion proteins (e.g., accessory colonization factors [*CBPs*], *InlJ*, and *Lap*). These genes support persistent intestinal colonization and biofilm formation, which are key traits for antagonizing pathogen colonization. Furthermore, sixteen genes encoded secretion systems (e.g., *Dot*/*Icm*, *T7SS*, *TTSS*, *T4SS*), potentially facilitating *Bifidobacterium* cross-feeding and metabolite exchange with gut commensal rather than pathogenic invasion or toxin production. Moreover, five to six associated genes included the bifunctional Cya, which may modulate host immune responses via cAMP regulation to exert anti-inflammatory effects. Among the 35 nonspecific virulence-associated genes, most were linked to iron and magnesium acquisition systems, suggesting these mechanisms may help the strain compete over intestinal resources while suppressing pathogen proliferation. The seven virulence-associated regulatory genes comprised three classes: BfmRS, RelA, and PhoP. Previous studies demonstrate that RelA and PhoP coordinate virulence factor expression with primary/secondary metabolite biosynthesis, ultimately affecting pathogenicity.

### 3.7. The Drug-Resistant B. animalis Strain’s Genotype and Phenotype Analysis

The antimicrobial phenotype of the *B. animalis* strain was assessed by the Kirby–Bauer disk diffusion method, which indicated 9 antibiotics of 7 categories, including quinolones, β-lactam, macrolides, amide alcohol, tetracyclines, aminoglycosides, and lincosamide. Furthermore, it was observed that the *B. animalis* strain was resistant to aminoglycosides (gentamicin) and β-lactam (ampicillin) while sensitive to β-lactam (ceftriaxone, penicillin), macrolides (tetracycline, erythromycin), amide alcohol antibiotics (chloramphenicol), quinolones (ciprofloxacin), and lincosamide (lincomycin). In β-lactam antibiotics, the strain was sensitive to ceftriaxone and penicillin while resistant to ampicillin. However, the BLAST analysis revealed that the whole genome of the *B. animalis* strain genome had 157 drug resistance genes in 24 categories based on the CARD (Appendix A, Figure 9). Through analysis, it has been found that there are several resistance genes in the genome, including *novA*, *mecC*, *tetB (60)*, *vanI, and PatB*. These genes may give the strain resistance to antibiotics such as neomycin, β—lactams, tetracyclines, vancomycin, and fluoroquinolones. In particular, PatB can interact with *PatA* to confer resistance to fluoroquinolone drugs. However, the analysis failed to detect the presence of the *PatA* gene. Although the strain harbored tetracycline resistance genes [*tetT*, *tetW*, *tetB(60)*, *tetA(58)*, *tetB(P)*], it demonstrated phenotypic sensitivity to this antibiotic, revealing a genotype–phenotype discordance. Furthermore, the strain genome also indicated various resistance and complex genes of different antibiotics, such as multiple pleuromutilin, fluoroquinolone, phenicol, streptogramin, oxazolidinone, aminocoumarin, rifamycin, penam, nitroimidazole, peptide, mupirocin, carbapenem, aminoglycoside, glycopeptide, monobactam, cephalosporin, acridine dye, cephamycin, diaminopyrimidine, isoniazid, and fosfomycin. These data revealed that *B. animalis* strain’s resistance mechanism might be complex. Furthermore, various drug-resistant phenotypes without specific resistance genes indicated the presence of different drug metabolism pathways.

### 3.8. Predictive Analysis of Pathogen–Host Interaction Between the B. animalis Strain and Host

The Pathogen–Host Interaction database annotation identified 291 genes in the *B. animalis* strain associated with pathogen–host interactions (Appendix A, Figure 10), including 199 reduced virulence, 30 losses of pathogenicity, 26 hypervirulence, 7 lethal factors, 7 effectors, 2 chemical resistances, 2 chemical sensitivities, while 82 genes did not affect pathogenicity. The effector and hypervirulence genes were the key genes that correlated with pathogenicity. The *B. animalis* strain hypervirulence genes included oligopeptide permeases (*App*), glutamine synthetase (*Rv2220*), two-component regulator system genes (*TcrX*/*Y*), probable aminopeptidase (*PepN*), virulence transcriptional regulatory factors (*Rv3167c* and Rho), 3-oxoacyl-ACP reductase (*gigX4*) genes, and ferrous ion transporters protein (*feoB*), whereas the effector genes included effector proteins (*map*, *LpdA*). These results indicated that only a few genes interact with the host and cause disease.

## 4. Discussion

Members of the *Bifidobacterium* genus are widely distributed in the gut microorganisms of humans and animals and exert beneficial symbiotic effects on their hosts. *Bifidobacterium* species prominently dominate the intestinal microorganisms of wild pigs (*Sus scrofa ussuricus*) more than domestic pigs [7,42,43], and their health-promoting activities have been documented across diverse animal hosts [44]. Despite their ecological significance, comprehensive genome analyses of *Bifidobacterium* strains isolated from wild animals, particularly wild pigs, remain limited. Therefore, this study isolated a novel *Bifidobacterium* strain from fresh wild pig feces using selective NPNL medium and analysis. The complete genome was sequenced via PacBio third-generation sequencing and polished with Illumina second-generation short-read data. Functional annotation revealed key genome features, including CRISPR-Cas systems and carbohydrate metabolism pathways, which were systematically analyzed to assess metabolic and adaptive capabilities. Furthermore, virulence-associated genes and antibiotic resistance profiles were characterized. This study not only provides a culture-based characterization of the isolated strain but also delivers a high-quality genome resource, enabling future investigations on the interplay between *Bifidobacterium* strains and host-specific traits in wild ecosystems.

The CRISPR-Cas system, a critical adaptive immune mechanism in bacteria, confers resistance against foreign DNA invasion, including plasmids and bacteriophages. This system has been extensively utilized in gene editing and genotyping due to its precision and versatility [45]. This study identified two CRISPR-Cas systems (CRISPR-Cas1 and CRISPR-Cas2) in the *B. animalis* genome (Appendix A), both harboring conserved Cas proteins essential for their function. These systems serve as key components of bacterial immunity against viral infections [46]. As a probiotic, the survival of *B. animalis* might be challenging in the harsh gastrointestinal tract environment, which harbors a dense reservoir of bacteriophages [47]. While temperate phages are prevalent in the gut [48], their presence threatens the viability of intestinal probiotics. CRISPR-Cas systems, in conjunction with Cas enzymes, are the primary defense mechanism employed by bacteria to counteract bacteriophage predation [49]. CRISPR-Cas systems confer bacterial immunity by integrating spacer sequences derived from invasive genetic elements, such as bacteriophages. In the *B. animalis* strain, 23 unique spacer sequences (29–39 bp in length) were identified within CRISPR loci, suggesting historical encounters with phage-derived DNA. However, despite comprehensive screening, no prophages were detected in the genome, indicating efficient clearance of integrated viral DNA. A higher spacer count in CRISPR-Cas systems typically correlates with enhanced immunity against diverse phages. Furthermore, spacer sequences showed no homology to residual prophage regions in this strain, indicating robust CRISPR-mediated targeting of exogenous invaders rather than self-genetic material. This absence of self-targeting underscores the system’s specificity and adaptive precision. Moreover, spacer variability provides a molecular basis for genotyping, as mismatches between spacers and extant prophages reflect active CRISPR loci countering foreign DNA. These indicate the possible defense mechanisms of *B. animalis* strain against phage predation and provide a basis for leveraging its CRISPR-Cas systems in genetic engineering applications.

The members of *Bifidobacterium* can synthesize and digest various carbohydrates [50]. Comparison analysis of CAZy databases showed that *B. animalis* strain had 78 potential carbohydrate-active, enzyme-related genes (Appendix A, Figure 7). CAZyme is the glycosyl hydrolase (GH) family, which catalyzes the hydrolysis of glycosidic linkages in various carbohydrate substrates. They can be classified as exo-or endo-glycoside hydrolases based on where they cleave the carbohydrate substrate [51]. It was observed that α-amylases (GH13), β-xylosidases (GH43), β-galactosidases (GH2, GH42, GH36), and β-glucosidases (GH1, GH3) were the most common polysaccharide utilization gene clusters-encoded degradation enzymes, which break down cellulose, oligosaccharides, and starch. Moreover, at polysaccharide utilization loci, genes encoding hemicellulosic polysaccharides and hydrolyzing enzymes (GH28, GH32) were frequently observed. It was also observed that *B. animalis* strains were primarily enriched with GH families, which degrade plant-derived carbohydrates, consistent with the plant-based wild pig’s diet [7]. Glycosyl hydrolase enzymes break down carbohydrates and belong to the subspecies-specific core genes. The GH13 family includes enzymes that hydrolyze poly- or oligosaccharides with α-glucosidic linkages, such as starch, glycogen, and related substrates, which might be because glycans can be found in the wild pig’s diet. The prediction of *B. animalis* species glycobiome revealed multiple GH13 enzymes, including pullulanase, cyclomaltodextrin glucanotransferase, trehalose-6-phosphate hydrolase, amylase, oligo-alpha-glucosidase, cyclomaltodextrinase, maltogenic amylase, neopullulanase, etc. These data indicated that this species could grow significantly on certain plant-derived carbohydrates, which was further validated by the fermentation ability analyses of various *B. animalis* taxon members. The results indicated that these species prefer utilizing different response-containing sugars (mannose, galactose, and glucose), and plant-derived glycans typically found in the wild pig’s diet, such as starch. Of the 7 *B. animalis*-specific GH enzymes, 1 belongs to the GH2 family. The GH2 family has exo-β-glucosaminidase [52] and galactosidase [53] activities, validating the observed increased galactose- and glucose-containing sugar metabolism of this taxon. The in silico analyses indicated many GH-encoding genes in the *B. animalis* strain, which further confirmed the identified broader carbohydrate-dependent growth activity.

Glycosyl-transferases use activated donors and transfer sugars to specific receptors such as lipids, proteins, or other glycans to form glycosidic linkages. The GT enzymes catalyze the glycosides formation involved in oligosaccharides, glycoconjugates, and polysaccharides biosynthesis [54] and have been associated with exopolysaccharide production in various bacterial species [55]. Of the 8 predicted GT families in the genome *B. animalis* strain, the GT2 family (Glycostransfase) was predominantly enriched. The CEs eliminate the glycan chain’s ester substituents to allow access to other CAZyme families in the carbohydrate chains [51]. Furthermore, they release *AxE* or *PON2* groups attached to carbohydrates via ester linkage [56], as well as carbohydrate-binding modules, which lack hydrolytic activity but interact with carbohydrate ligands and elevate the carbohydrate-active enzyme’s catalytic efficiency [56]. Because of the large size and increased polymerization of mucin glycans, their degradation starts extracellularly. Therefore, some GH enzymes are located on the environment or are present on the cell surface [57]. Bacterial GHs are essentially involved in the degradation of mucin glycan. Antibiotics use restrictions in pigs has increased the search for novel probiotics as an alternative. *Bifidobacterium* species that colonize animal guts have received much attention because they provide health benefits to their host, such as supporting gut microbial ecosystem development by producing short-chain fatty acids, promoting colonization resistance against pathogens, and modulating immunity. Furthermore, they also promote the competitive exclusion of pathogens [58,59], degrade diet-derived carbohydrates [60], and modulate the immune system [61,62]. Therefore, they are often added to probiotic products in combination with other lactic acid bacteria to prevent or treat diseases [63,64]. Meanwhile, BALB/c mice were selected as experiment animal, and a mouse disease model was constructed to explore the probiotic effects of *B. animalis* and its preparations in vivo. The results showed that ETEC infection caused a trend of weight loss in mice, severe damage to the morphological of intestinal tissue, shortening of intestinal villi, increased crypt depth, destruction of intestinal barrier function, increased intestinal permeability, and the induction of intestinal inflammation. Feeding animals with *B. animalis* probiotics and its preparations could improve the immune organ index, up-regulate the levels of anti-inflammatory cytokines *IL-4* and *IL-10* in serum, and inhibit the expression levels of pro-inflammatory cytokines *IL-6* and *TNFα*, so as to reverse the damage of ETEC to intestinal tissue as a way of enhancing the immune capacity of the organism (Results are pending publication). These data indicated that complex plant polysaccharides and host glycans degradation in wild pigs might be linked with the enzymatic synergy of various uncultivated microorganisms, as well as indicating that *B. animalis* has disease-resistant effects.

BLAST analysis against the CARD revealed 157 antibiotic resistance genes in the complete genome of the *B. animalis* strain (Appendix A, Figure 9). Furthermore, there was a significant presence of tetracycline-associated resistance genes (*tetT*, *tetW*, *tetB(60)*, *tetA(58)*, and *tetB(P)*). Among Bifidobacteria, tet genes serve as the primary genetic determinant of tetracycline resistance, with *tet(W)* exhibiting the highest detection frequency in *B. animalis* [65,66,67,68]. However, despite harboring these resistance genes, the *B. animalis* strain indicated a phenotypic susceptibility to tetracycline, indicating genotype–phenotype discordance, a finding consistent with previous observations [69]. Antibiotic susceptibility profiles varied significantly across *Bifidobacterium* species. While strain-specific susceptibility to ciprofloxacin has been documented [70,71], tetracycline resistance remains the most prevalent phenotype in this genus [65,66,70]. The results suggest that the presence of resistance genes signifies potential rather than expressed resistance. These genes may either remain transcriptionally inactive or require synergistic interactions with auxiliary genetic elements to manifest phenotypic resistance, underscoring the need for functional validation of genotype–phenotype correlations. Furthermore, genomic analysis confirmed the absence of plasmid structures in the isolated *B. animalis* strain. The lack of plasmid-borne resistance genes markedly reduces the likelihood of horizontal gene transfer to pathogenic bacteria, thus diminishing the risks of antimicrobial resistance dissemination.

This research also identified 104 putative virulence genes in the *B. animalis* strain via the VFDB analysis (Appendix A, Figure 8). This investigation is the first to report the virulence factors of the *B. animalis* strain. Several studies have confirmed the antibacterial and antiviral activities of probiotics, including the *B. animalis* strain [12]. However, the mechanisms of these effects were investigated mostly in vitro. These studies revealed that various genes were associated with bacterial virulence, such as genes encoding proteins involved in adherence (*Flagella*, *Flp type IV pili*, *CBPs*, *InlJ*, *Lap*, *LPS*, *Hsp60*, *IlpA*), serum resistance (*LPS*), anti-phagocytosis (Alginate regulation, Capsule), stress protein (*ClpC*, *MsrAB*, *ClpP*), secretion systems (*Dot/Icm*, *T7SS*, *TTSS*, *T4SS*, *HSI-I*), iron uptake (*HitABC*, *FbpABC*, *FupA*), and toxin (*Cya*), which is a risk factor of infection. The results of this research should be cautiously interpreted because these virulence factors are also crucial features of most commensals. Furthermore, most *Bifidobacteria*-to-host tissue adhesion mechanisms are similar and even identical to those utilized by disease-causing pathogens [72]. Moreover, limited variations were observed in the putative virulence gene content, and most genes were found in both noninvasive and invasive isolates. The VFDB genome analysis indicated 8 adherence mechanisms, 3 capsule formations, and 2 biofilm formation-related proteins. These genes were also observed in the reference strains submitted to the website for comparison. However, strains carrying virulence genes do not always indicate pathogenicity or cause disease. Therefore, these genes might be essentially associated with bacterial colonization and proliferation in the gastrointestinal tract, a crucial probiotic property to fight pathogenic bacteria. Altogether, the data indicated that the *B. animalis* strain may have a reduced pathogenicity; therefore, it may be innocuous to the host.

Although this study characterized the biological properties and genomic functions of a wild boar-derived *B. animalis* strain. However, the characterization of the *B. animalis* strain lacks in vivo animal validation. First, the current research solely focused on a single wild pigs isolate, limiting its representativeness, and did not systematically compare this strain’s genomic and functional characteristics with commercial or domestic pig-derived strains, weakening the basis for evaluating its practical application potential. Second, functional analysis primarily relied on genomic predictions and in vitro phenotypes, lacking validation through in vivo experiments to confirm predicted gene functions in the host environment or elucidate molecular mechanisms underlying genotype–phenotype discrepancies. These limitations highlight the need for future research integrating in vivo validation and comparative genomics.

## 5. Conclusions

This study successfully isolated a *B. animalis* strain from the fresh feces of wild pigs. The *B. animalis* strain had certain acid resistance, bile salt resistance, gastrointestinal fluid tolerance, and antibacterial characteristics. Whole genome sequencing revealed that the strain harbors diverse carbohydrate-active enzyme (CAZyme) genes, which are implicated in specialized metabolic pathways for degrading dietary and host-derived intestinal substrates, thereby contributing to gut homeostasis. Although the genome indicated antibiotic resistance genes, no direct correlation was observed with phenotypic resistance. Furthermore, the low pathogenicity potential of virulence-associated genes supports *B. animalis* strain safety as a probiotic candidate. These findings provide molecular insights into the protective role of *Bifidobacterium* in pig’s gut ecosystems and expand the repository of symbiotic microbial resources from wild mammals. This study provided a foundation for developing host-adapted probiotics and offers critical genomic data for future studies on Bifidobacterial ecology and applications.

## Figures and Tables

**Figure 1 microorganisms-13-01666-f001:**
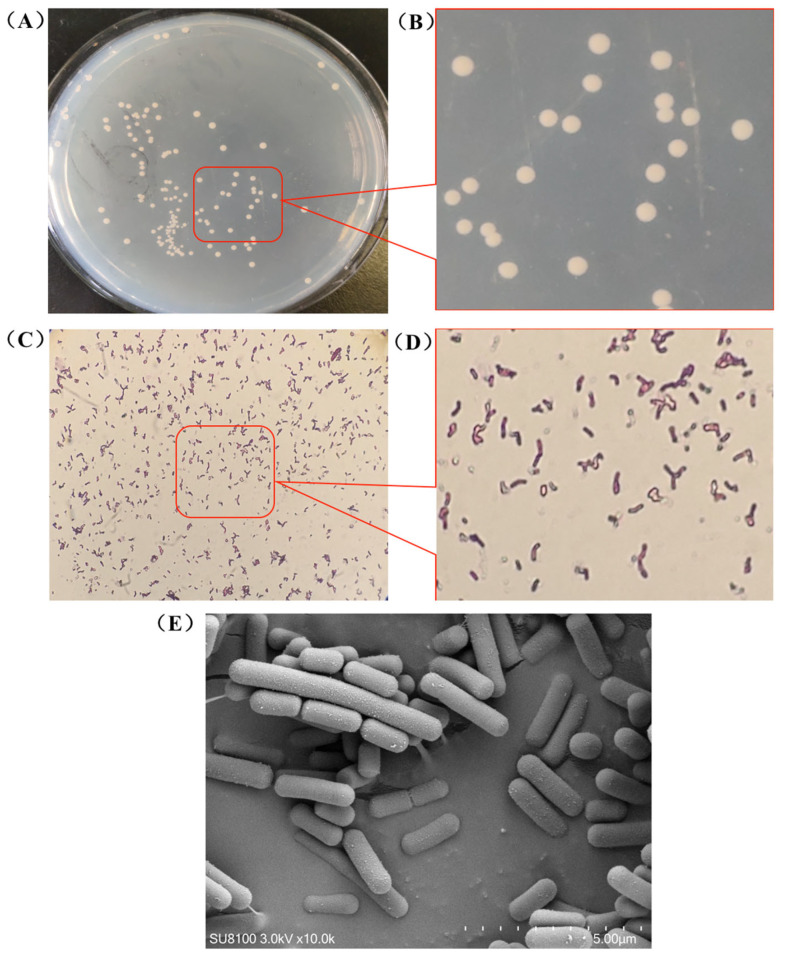
Isolation and culture morphology of *Bifidobacterium* strains (**A**,**B**), Gram staining (10 × 40 μm) (**C**,**D**), and scanning electron microscopy (10,000 × 5 μm) (**E**).

**Figure 2 microorganisms-13-01666-f002:**
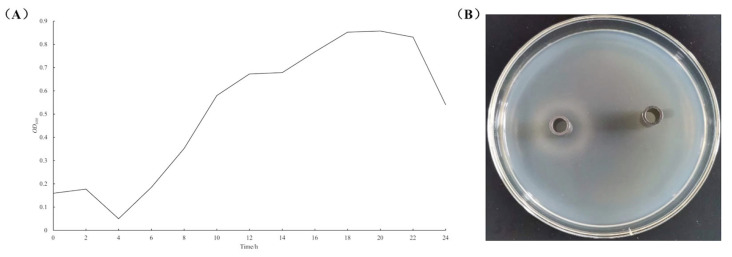
Growth curve (**A**) and in vitro antibacterial assay (**B**) of the *B. animalis strain*.

**Figure 3 microorganisms-13-01666-f003:**
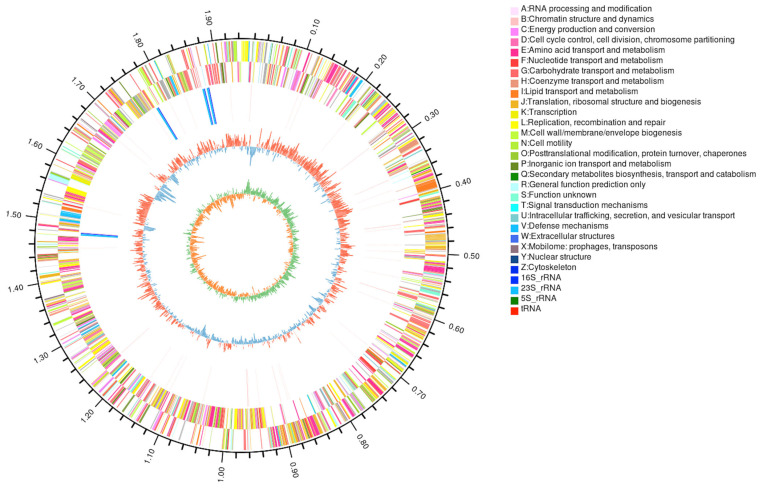
Circular genome maps of the *B. animalis* strain.

**Figure 4 microorganisms-13-01666-f004:**
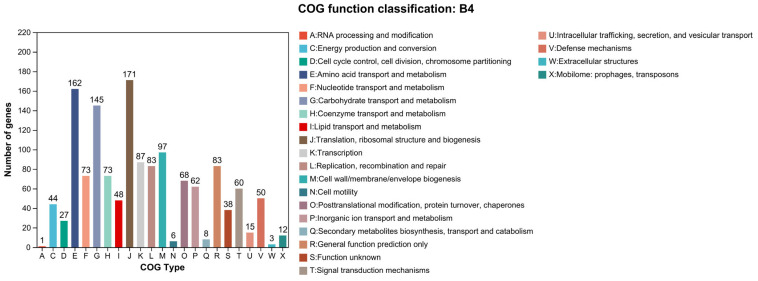
COG functional annotation of the *B. animalis* strain.

**Figure 5 microorganisms-13-01666-f005:**
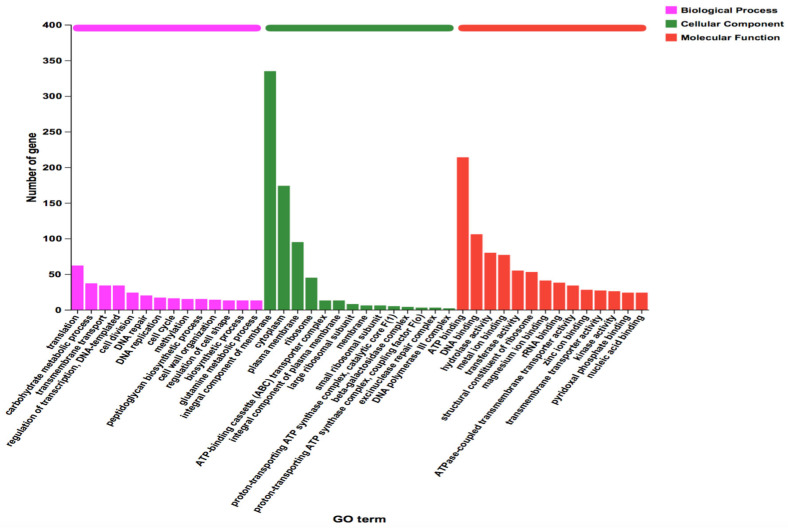
GO functional annotation CDS in the whole genome of the *B. animalis* strain.

**Figure 6 microorganisms-13-01666-f006:**
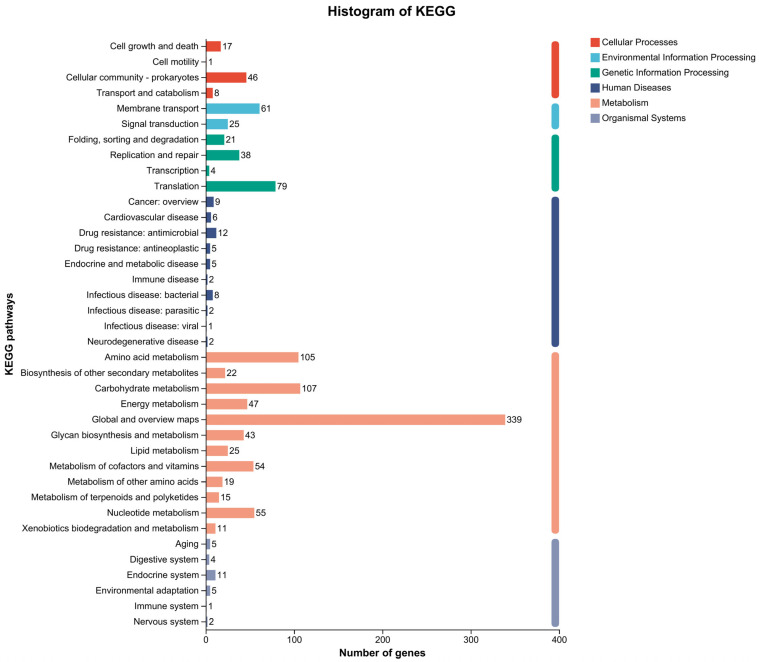
KEGG pathway annotation of the *B. animalis* strain. The ordinate indicates the level 2 KEGG pathway classification, and the abscissa indicates the number of genes under the annotation of this classification. Different column colors represent the level 1 KEGG pathway classification.

**Figure 7 microorganisms-13-01666-f007:**
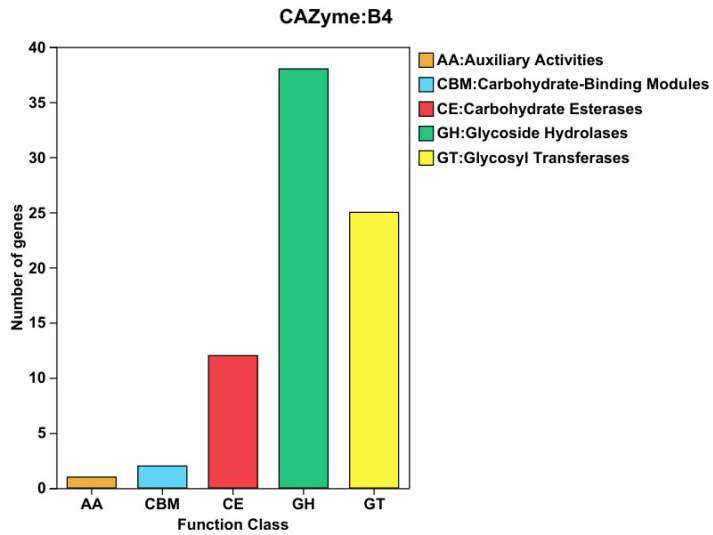
CAZy annotation statistics chart. Note: Colors represent categories, and the area represents the proportion of genes in each category.

**Figure 8 microorganisms-13-01666-f008:**
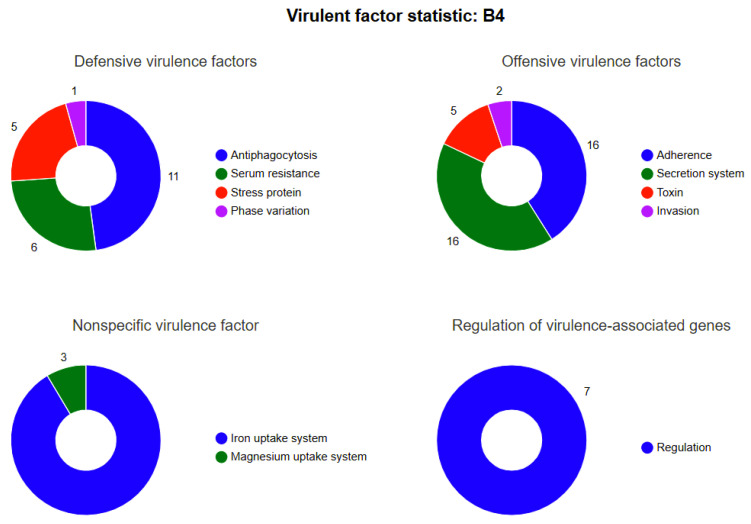
Schematic representation of virulence factor prediction. Note: The inner circle labels indicate primary classifications of virulence factors, while the OTUer circle annotations designate secondary subcategories. Color gradients correspond to distinct secondary classifications, with proportional sectors representing the relative abundance of genes within each subcategory.

**Figure 9 microorganisms-13-01666-f009:**
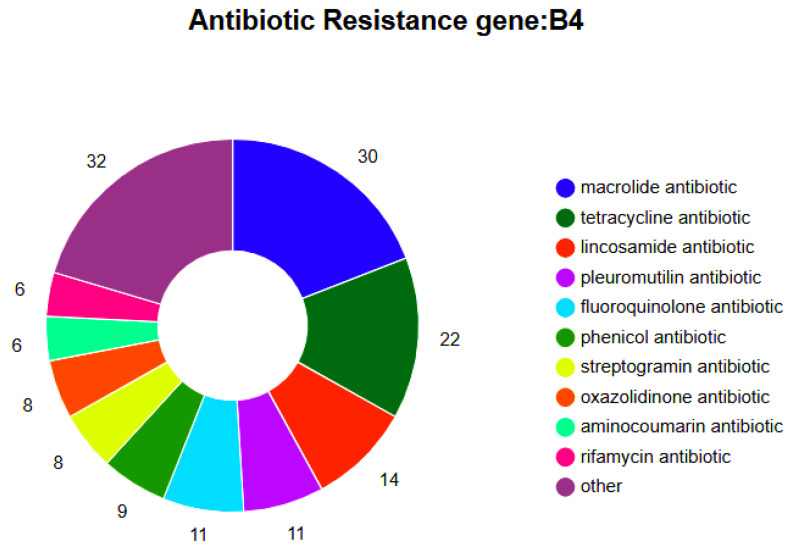
Composition of antimicrobial resistance genes. Note: Color coding denotes distinct antimicrobial categories, with sector proportions reflecting the percentage distribution of resistance genes within each drug class.

**Figure 10 microorganisms-13-01666-f010:**
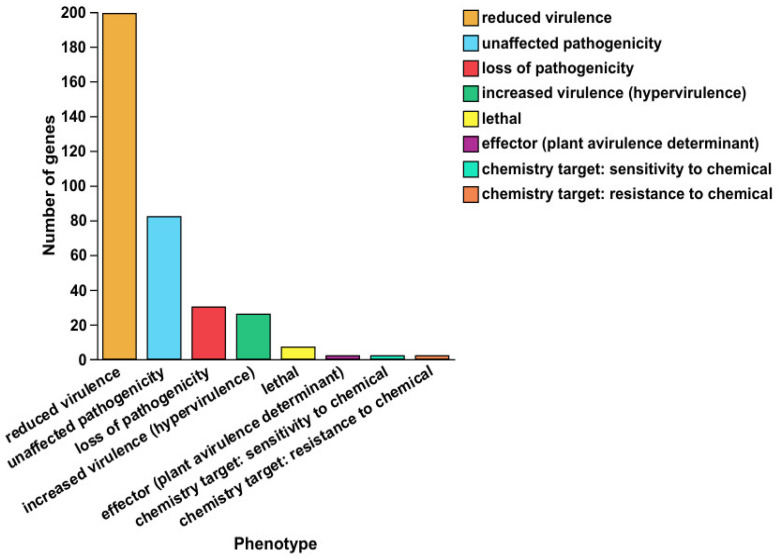
Bar chart of pathogen-predicted genes. Note: The horizontal axis denotes phenotypic categories; the vertical axis denotes the number of genes per category.

## Data Availability

The whole genome sequence of the *Bifidobacterium animalis* strain has been deposited into the GSA database of the National Data Centre for Genome Sciences (Project No. PRJCA028163).

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
