# Peer review of "Complete Genome and Characterization Analysis of a Bifidobacterium animalis Strain Isolated from Wild Pigs (Sus scrofa ussuricus)"

_microorganisms, 2025, doi:10.3390/microorganisms13071666_

Round 1
Reviewer 1 Report
Comments and Suggestions for Authors
The title should be: Characterization and Complete Genome Analysis of a Bifidobacterium animalis Strain Isolated from Wild Pigs (Sus scrofa ussuricus)
Line 97: What broths were used? Why NPNL medium?
Line 123: Maybe the sentence is better with the removal of the words “for 4 treatments”.
Lines 46, 178, 206, 273, 292, 346: Streptococcus and Lactobacillus spp., Bifidobacterium or B. animalis should be written in italics. Please, check carefully the entire manuscript.}
Lactobacillus is now a genera include in lactobacilli family (there is a new nomenclature). So, I suggest to refer to the lactobacilli family.
Although the strain was genetically analyzed the characterization must be complemented through in vitro and in vivo experiments (immunological effect and inhibitory effect against pathogens).
The reading of the information in the figures it´s difficult, because of the size of the letters.
Author Response
Comments 1: Line 97: What broths were used? Why NPNL medium?
Response 1: Thank you for pointing this out. We agree with this comment and apologies for our negligence. Firstly, we shall explain that TPY liquid medium is a culture medium for Bifidobacterium proliferation. Modified NPNL selective medium, containing neomycin, baromycin, nadifloxacin and lithium chloride, is used for the selective separation and culture Bifidobacteria. It can inhibit other bacteria. Therefore, we have changed in line 98 of the original text, the specific names of the culture broths are added as TPY medium.
Comments 2: Lines 97:Maybe the sentence is better with the removal of the words “for 4 treatments”.
Response 2: Agree. As suggested, we have deleted the words “for 4 treatments” in lines 126.
Comments 3: Lines 46, 178, 206, 273, 292, 346: Streptococcus and Lactobacillus spp., Bifidobacterium or B. animalis should be written in italics. Please, check carefully the entire manuscript.}
Response 3: Our apologies for this negligence. We agree with this comment. As suggested, the Streptococcus and Lactobacilli family, Bifidobacterium or B. animalis have been changed italics in entire manuscript.
Comments 4: Lactobacillus is now a genera include in lactobacilli family (there is a new nomenclature). So, I suggest to refer to the lactobacilli family.
Response 4: Thank you for your valuable suggestions. Regarding the updated nomenclature for Lactobacillus you mentioned, W have fully revised Lactobacillus to Lactobacilli family according to new naming conventions.
Comments 5: Although the strain was genetically analyzed the characterization must be complemented through in vitro and in vivo experiments (immunological effect and inhibitory effect against pathogens).
Response 5: Thank you for your useful suggestions. We've conducted in vitro experiments to study B.animalis antibacterial assay in lines 147-158 and results in lines 210-213. These experiments will offer more comprehensive results.
Comments 6: The reading of the information in the figures it´s difficult, because of the size of the letters.
Response 6: We apologize for the oversight. We have increased the font size as much as possible to enhance clarity and readability.
Response to Comments on the Quality of English Language
Response 1: MJEditor (www.mjeditor.com) have provided english language editing services during the preparation of this manuscript.

Reviewer 2 Report
Comments and Suggestions for Authors
The manuscript presents the isolation, phenotypic characterisation and complete genomic analysis of a Bifidobacterium animalis strain. The authors describe the isolation protocol, identification by 16S rRNA sequencing and finally the detailed genomic analysis. The work is methodologically sound in the experimental part, but the description of the results and interpretations has some unclear or insufficiently detailed areas, especially concerning the criteria for strain selection, experimental controls, functional implications of genomic annotations and the relationship between phenotypic and genotypic data. There are also some gaps in the discussion of study limitations and application perspectives.
- In the manuscript in some cases the names of classes of bacteria are indicated in italics while in other cases they are not, please standardise.
- Line 74, I advise the authors to indicate the full name of the abbreviation (B. animalis) also in the text, although it is included in the abstract.
- I would advise the authors to add a selection in Materials and Methods where they describe the protocols used for the various microscopic analyses as they are not present in the text.
- I do not understand why data that are discussed in the text are present in supplementary material. For example, paragraph 3.2 describes images and data that are not present in the final text, I would recommend including it.
- Figure 2 is very beautiful but it is difficult to read. I would advise the authors to improve the quality because by enlarging it to better observe it, it completely loses quality and it becomes very difficult to understand the various differences.
- In the caption of fig. 2, abbreviations are included that are only explained later, making it very complex to understand. I advise the authors to correct this.
- In fig.4 there is a legend in the top right hand corner indicating the various colours to which they refer, I think it is pointless to rewrite it at the bottom.
- Likewise as before, paragraphs 3.5, 3.6, 3.7 and 3.8 are paragraphs without any data. I would advise the authors either to insert the text directly in MS, or to insert the tables in the full text.
- How many strains of B. animalis were isolated from faeces samples in total? Does the analysed strain represent a random choice or was it selected according to specific criteria (e.g. phenotype, growth, resistance)?
- What is the average coverage and depth of genome sequencing obtained for this strain? Were any uncovered or problematic genomic regions found in the assembly?
- In morphological characterisation, were differences observed compared to reference strains of B. animalis? If so, what implications might these differences have?
- Genomic annotation identified antibiotic resistance genes: what exactly are these genes and to which classes of antibiotics do they confer potential resistance?
- When comparing phenotypic and genotypic data on antibiotic resistance, what are the main discrepancies found and how are they interpreted?
- Has a direct comparison been made between the genome of this strain and that of other B. animalis strains isolated from domestic pigs or other animal species?
- I advise the authors to include a paragraph with the main limitations of the study in order to improve the overall quality of the manuscript.
Author Response
Comments 1: The manuscript presents the isolation, phenotypic characterisation, and complete genomic analysis of a Bifidobacterium animalis strain. The authors describe the isolation protocol, identification by 16S rRNA sequencing and finally the detailed genomic analysis. The work is methodologically sound in the experimental part, but the description of the results and interpretations has some unclear or insufficiently detailed areas, especially concerning the criteria for strain selection, experimental controls, functional implications of genomic annotations and the relationship between phenotypic and genotypic data. There are also some gaps in the discussion of study limitations and application perspectives.
Response 1: Thank you very much for your attention to our research and for your valuable suggestions. We agree with your suggestions. Regarding the issues you raised about the criteria for strain selection, experimental control, the functional significance of genome annotation, and the relationship between phenotypic and genotypic data, we have carefully considered these problems.
In terms of strain selection, our current research is based on specific experimental objectives and the resources available to us. For experimental control, we have added relevant experimental methods in the Materials and Methods section. In terms of genome annotation, we have provided a detailed explanation of the functional significance to understand the functional roles. However, some functions and the relationship between phenotypic and genotypic data have been preliminarily discussed but may require further in-depth research. Regarding the discussion of the limitations and future applications, we have briefly mentioned the limitations of the current research and provided an outlook on potential future applications in the text. We have also added paragraphs in the article to address these issues.
Comments 2: In the manuscript in some cases the names of classes of bacteria are indicated in italics while in other cases they are not, please standardise.
Response 2: We agree with your valuable suggestions, and we have correctted the non-standard italics for the bacterial names in the main text.
Comments 3: - Line 74, I advise the authors to indicate the full name of the abbreviation (B. animalis) also in the text, although it is included in the abstract.
Response 3: Thank you very much for your valuable suggestions. We agree with your recommendations. We have revised the full name of the abbreviation (B. animalis) in the text.
Comments 4: I would advise the authors to add a selection in Materials and Methods where they describe the protocols used for the various microscopic analyses as they are not present in the text.
Response 4: Thank you for your helpful suggestions. We have added “Microscopic analyses of the B. animalis” in Materials and Methods where they describe the protocols used for the various microscopic analyses.
Comments 5: I do not understand why data that are discussed in the text are present in supplementary material. For example, paragraph 3.2 describes images and data that are not present in the final text, I would recommend including it.
Response 5: This was an oversight on our part. Thank you very much for pointing it out. We have now added the paragraph 3.2 describes images (Figure 2) in the main text.
Comments 6: Figure 2 is very beautiful but it is difficult to read. I would advise the authors to improve the quality because by enlarging it to better observe it, it completely loses quality and it becomes very difficult to understand the various differences.
Response 6: Thank you for your valuable feedback. We have improved the image quality of Figure 2 (now Figure 3) to make its content clearer and facilitate reading.
Comments 7: In the caption of fig. 2, abbreviations are included that are only explained later, making it very complex to understand. I advise the authors to correct this.
Response 7: Thank you for your valuable suggestions. As suggested, we have revised the annotations in Figure 2 (now Figure 3) of the main text based on your comments.
Comments 8: In fig.4 there is a legend in the top right hand corner indicating the various colours to which they refer, I think it is pointless to rewrite it at the bottom.
Response 8: Thank you for your valuable feedback. Regarding the legend in the upper right corner of Figure 4 (now Figure 5), we acknowledge your observation that replicating it at the bottom would be redundant. As suggested, we have rewrite it at the bottom.
Comments 9: Likewise as before, paragraphs 3.5, 3.6, 3.7 and 3.8 are paragraphs without any data. I would advise the authors either to insert the text directly in MS, or to insert the tables in the full text.
Response 9: Thank you for your suggestion. As suggested, we have incorporated your recommendation by inserting relevant data figures (Figure 7, Figure 8, Figure 9, Figure 10) after paragraphs 3.5, 3.6, 3.7, and 3.8 in the main text to correspond with the descriptions therein.
Comments 10: How many strains of B. animalis were isolated from faeces samples in total? Does the analysed strain represent a random choice or was it selected according to specific criteria (e.g. phenotype, growth, resistance)?
Response 10: We appreciate your suggestions. A total of five B. animalis strains were isolated from wild pigs feces, and one randomly selected strain was subjected to analysis according to specific criteria (e.g. phenotype, growth, resistance).
Comments 11: What is the average coverage and depth of genome sequencing obtained for this strain? Were any uncovered or problematic genomic regions found in the assembly?
Response 11: We appreciate your valuable feedback on our study. During sequencing analysis, while a depth of 1,000× was achieved, the sequencing provider did not report the average coverage. Nevertheless, K-mer frequency distribution analysis revealed no anomalies in genomic regions.
Comments 12: In morphological characterisation, were differences observed compared to reference strains of B. animalis? If so, what implications might these differences have?
Response 12: Thank you for your comments. We found no differences between our strain and other animal bifidobacterial strains in the analysis of morphological characterization.
Comments 13: Genomic annotation identified antibiotic resistance genes: what exactly are these genes and to which classes of antibiotics do they confer potential resistance?
Response 13: Thank you for your suggestions.We identified several resistance genes in the genome, including novA, mecC, tetB (60), vanI, and PatB. These may cause resistance to neomycin, β - lactams, tetracyclines, vancomycin, and fluoroquinolones. PatB can interact with PatA to confer fluoroquinolone resistance, but PatA wasn’t detected here. We revised this part in lines 376 - 382 in the manuscript.
Comments 14: When comparing phenotypic and genotypic data on antibiotic resistance, what are the main discrepancies found and how are they interpreted?
Response 14:Thank you for your suggestions. We identified tetracycline resistance genes (tetT, tetW, etc.) in the genome, but phenotypic assays showed susceptibility. This discordance—documented in other bifidobacteria—likely stems from unexpressed genes or missing auxiliary cellular components.
Comments 15: - Has a direct comparison been made between the genome of this strain and that of other B.animalis strains isolated from domestic pigs or other animal species?
Response 15: Thank you for your insightful comment. In current study, we aim to analysis the complete genome and characterization of this specific strain.We did not conduct a direct comparison of the genome of this strain with other Bifidobacterium strains isolated from domestic pigs or other animal species. Based on the question raised by the reviewers, and the current literature reports (Bottacini et al., 2011; Barrangou et al., 2009; Loquasto et al., 2011; Kang et al., 2017; et al., 2024; Tsukimi et al., 2020), there is indeed no comparative analysis of the composition between the genomes of B.animalis from different species, we apologize for our negligence. In the next step, we will download other species of the B.animalis genomes and conduct-depth comparative analysis to fully understand the genomic structural characteristics of the B.animalis.
Reference:
Bottacini, F., Dal Bello, F., Turroni, F., et al. 2011. Complete genome sequence of Bifidobacterium animalis subsp. lactis BLC1.Journal of Bacteriology, 2011,193(22):6387–6388.
Barrangou, R., Briczinski, E.P., Traeger, L.L.,et al. Comparison of the complete genome sequences of Bifidobacterium animalis subsp. lactis DSM 10140 and Bl-04. Journal of Bacteriology, 2009,191(13):4144–4151.
Loquasto, J.R., Barrangou, R., Dudley, E.G. et al. The complete genome sequence of Bifidobacterium animalis subspecies animalis ATCC 25527T and comparative analysis of growth in milk with B. animalis subspecies lactis DSM 10140T. Journal of Dairy Science, 2011, 94(12): 5864–5870.
Kang, J., Chung, W.H., Lim, T.J., et al. Complete genome sequence of the Bifidobacterium animalis subspecies lactis BL3, preventive probiotics for acute colitis and colon cancer. New Microbes and New Infections, 2017, 19:34-37.
Li, J., Wu, X., Ding, B. et al. Complete genome sequence of Bifidobacterium animalis B01. Microbiology Resource Announcements, 2024, 13(8):e00394-24.
Tsukimi, T., Watabe, T., Tanaka, K., et al. Draft genome sequences of Bifidobacterium animalis consecutively isolated from healthy japanese individuals. Journal of Genomics, 2020, 8:37–42.
Comments 16: I advise the authors to include a paragraph with the main limitations of the study in order to improve the overall quality of the manuscript.
Response 16: Thank you for your insightful comment.As suggested, we have added a paragraph with the main limitations in the manuscript in lines 567-577.
Response to Comments on the Quality of English Language
Response 1: MJEditor (www.mjeditor.com) have provided english language editing services during the preparation of this manuscript.
